# Southward expanding plate coupling due to variation in sediment subduction as a cause of Andean growth

Jiashun Hu [1,2,3 ✉], Lijun Liu[3] & Michael Gurnis[2]

Growth of the Andes has been attributed to Cenozoic subduction. Although climatic and tectonic processes have been proposed to be first-order mechanisms, their interaction and respective contributions remain largely unclear. Here, we apply three-dimensional, fully-dynamic subduction models to investigate the effect of trench-axial sediment transport and subduction on Andean growth, a mechanism that involves both climatic and tectonic processes. We find that the thickness of trench-fill sediments, a proxy of plate coupling (with less sediments causing stronger coupling), exerts an important influence on the pattern of crustal shortening along the Andes. The southward migrating Juan Fernandez Ridge acts as a barrier to the northward flowing trench sediments, thus expanding the zone of plate coupling southward through time. Consequently, the predicted history of Andean shortening is consistent with observations. Southward expanding crustal shortening matches the kinematic history of inferred compression. These results demonstrate the importance of climate-tectonic interaction on mountain building.

[1] Department of Earth and Space Sciences, Southern University of Science and Technology, Shenzhen 518055, China. [2] Seismological Laboratory, California Institute of Technology, Pasadena, CA 91125, USA. [3] Department of Geology, University of Illinois at Urbana-Champaign, Champaign, IL 61820, USA. ✉email: hujs@sustech.edu.cn

The magnitude of Cenozoic Andean crustal shortening varies significantly along the margin[1–3]. The largest shortening occurred within the middle of the convergent margin from around 10°S to 33°S and tapered out towards the north and south, fostering Andean growth since the Eocene[4] (Fig. 1a). Various mechanisms have been proposed to account for observed Andean shortening, which generally falls into two categories, tectonic or climatic. Tectonic mechanisms highlight the effects of the subducting plate age[5], slab dip angle[6,7], the thickness and westward motion of the overriding plate[5,8], slab–lower mantle interaction[9], or slab-induced mantle flow[10–12], while climatic mechanisms emphasize coupling along the plate interface due to sediment subduction[13,14] or surface erosion[15,16].

Several key observations related to Andean shortening have been noted earlier. First, the present-day trench segment from 12°S to 33.5°S is starved of sediments (Fig. 1c). Seismic reflection surveys have revealed a sediment thickness of ≤0.5 km within the central Andean trench, in contrast to thicknesses of 1–3 km in northern and southern Andes[17,18]. The present-day sediment thickness has a largely inverse correlation with Andean topography and the width of the orogen[13] (Fig. 1c). It has been suggested that sediments could lubricate or weaken the plate interface[13,14]. Sediments in the subduction channel have low frictional resistance and could release fluids and elevate pore pressure, which reduces the mechanical strength along the plate boundary faults[19]. Additionally, the effective viscosity of

sediments is 2–3 orders of magnitude lower than subducted mafic rocks[20], and the blanketing of the oceanic basement by sediments could warm the subducted plate[13], both reducing the viscous coupling between the two plates. Conversely, trench sediment depletion may increase the strength of plate coupling, which then enhances deformation within the upper plate (see discussions in Supplementary Information).

Second, the transition from sediment enrichment to sediment starvation along the trench is sharp and coincides with the location where the JFR intersects the coast of Chile at 33.5°S (Fig. 1c and Supplementary Fig. 1). This suggests the thick sediments south of the ridge are dammed by the topographic relief of JFR[18,21] (see discussions in Supplementary Information). The flat-lying trench sediments south of the ridge (Supplementary Fig. 1) suggest a mass redistribution of sediments within the trench[17,18,21]. In fact, a prominent trench axial channel has been observed continuously from 44°S to 31°S (Supplementary Fig. 1), interpreted as the pathway of turbidity currents inside the trench[21]. These observations suggest the sediments can travel thousands of kilometers within the trench[17,21] from south to north, following the descending direction of the trench topography.

Third, plate kinematics in regional and global reconstructions consistently suggests a southward migrating JFR (Fig. 1b). Assuming a fixed Juan Fernandez hotspot, the JFR hotspot track can be reconstructed using a recent plate reconstruction[22]. The

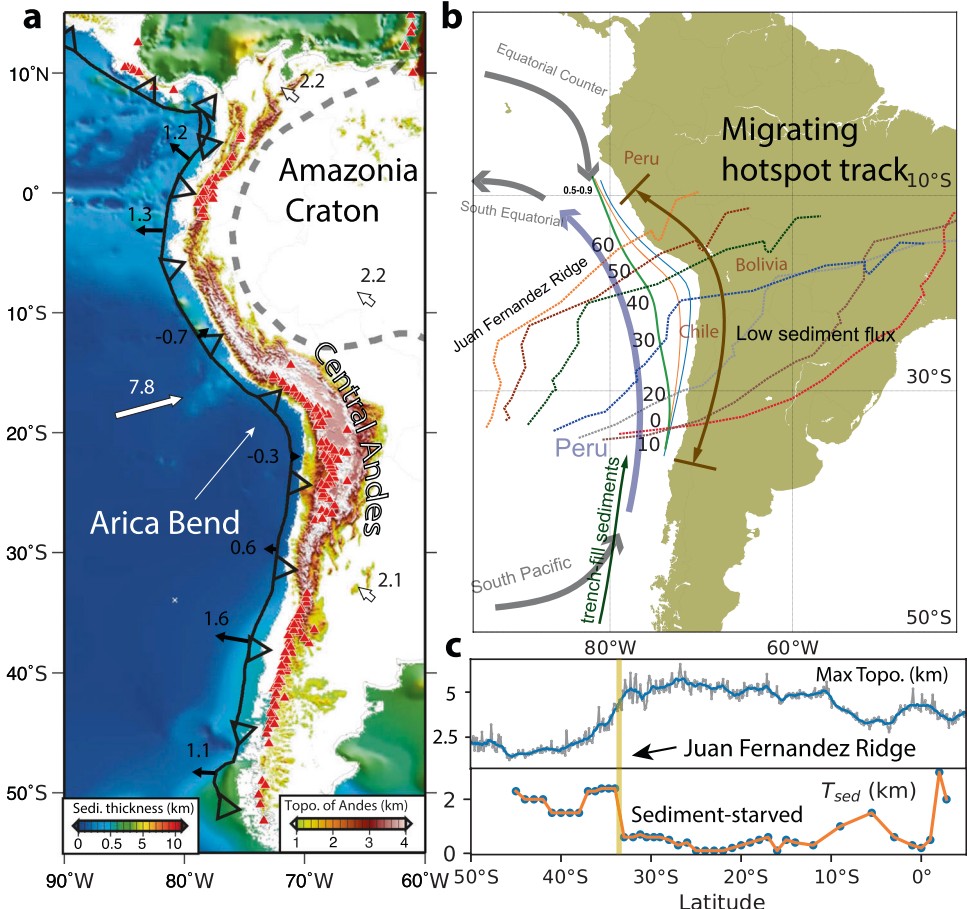

**Fig. 1 Trench-fill sediments and tectonics of South American subduction. a** Sediment thickness in oceans[58] and the topography of the Andes. White thick arrows indicate plate velocities. Black thin arrows indicate trench velocities. **b** Reconstruction of the Juan Fernandez Ridge relative to South America[22]. Dashed lines represent the paleo-positions of JFR from 60 Ma to present with an interval of 10 Myrs. Thick semi-transparent lines represent ocean currents, with the cold Humboldt Current (or Peru Current) colored with blue and the rest with black. The green arrow indicates the transport direction of trench-fill sediments. **c** variation of the maximum Andean topography and the thickness of trench-fill sediments with latitude.

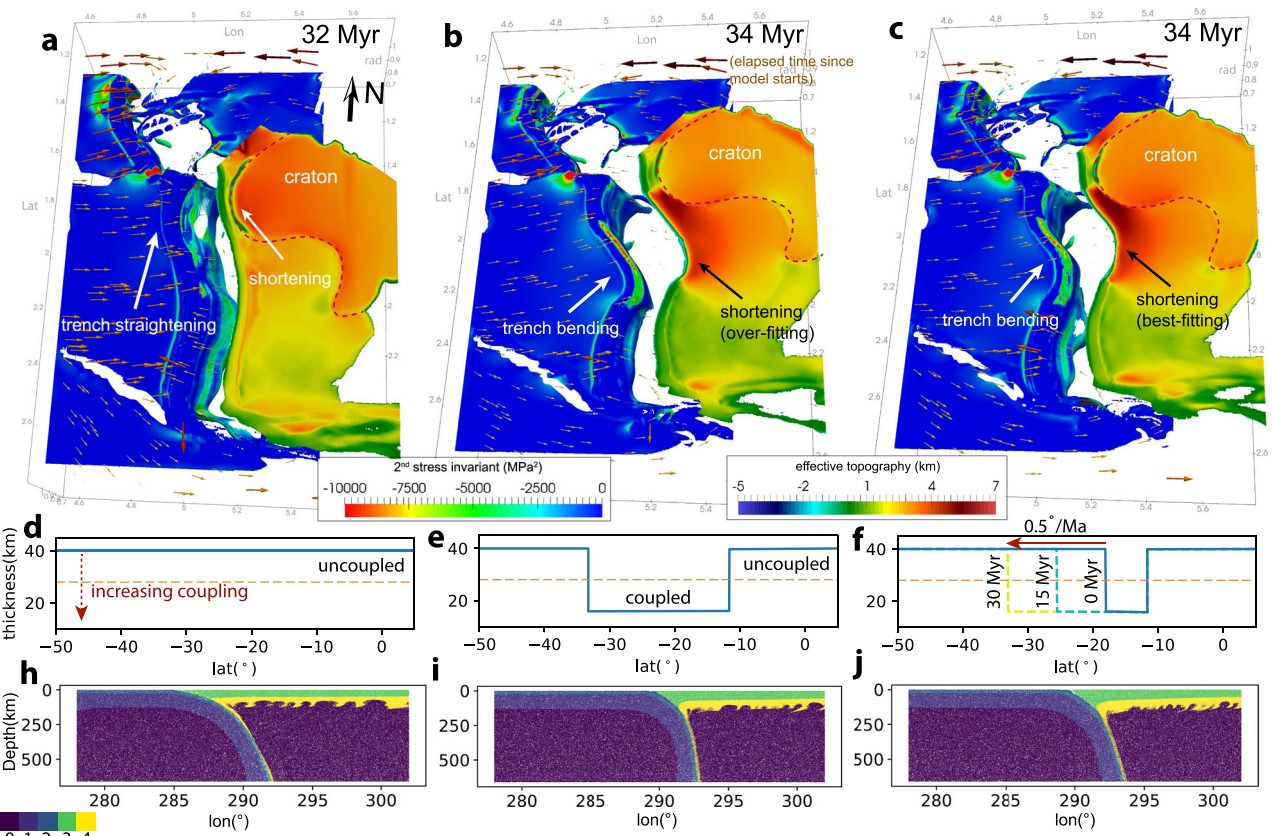

**Fig. 2 The final snapshots of numerical experiments on central Andean shortening. a** Model 1 where the lubricating layer has a uniform thickness (40 km) and the two plates are uncoupled along the trench. **b** Model 2 that has a plate coupling zone in the central Andes. The trench segment from 15°S to 35°S has a 20 km-thick lubricating layer, while the rest of the trench has a 40 km-thick lubricating layer. **c** Model 3 that has a southward expanding plate coupling zone, where the southern boundary of the plate coupling zone migrates towards the south with a rate of 0.5°/Myr. **d–f** Show the corresponding thickness of the lubricating layer in Model 1–3. **h–j** Show the cross-sections of the compositional fields at 22°S in Models 1–3. Compositional types 0–4 represent the ambient mantle, oceanic mantle lithosphere, oceanic crust, continental crust, and continental mantle lithosphere, respectively. Note crustal shortening in **i** and **j**. In **a–c**, the effective topography is shown on the South American Plate, which reflects crustal shortening shown in **h–j**, with reddish colors indicating stronger shortening than bluish colors. The second invariant of the stress tensor is shown on the oceanic plates. The coordinates in **a–c** are shown in radians.

predicted JFR intersected the trench at southern Peru in the middle Eocene (50–40 Ma). Since then the ridge migrated southward relative to the South American plate towards the present location of 33.5°S until 12 Ma then it has been relatively stationary. Several recent plate models[22–26] yield slightly different positions of JFR relative to South America (see the "Methods" section), but they all agree on the overall southward migration. This is consistent with independent evidence on tectonic deformation[27] and magmatic arc migration[28] within central Chile.

In this study, we propose trench-axial sediment transport and subduction is an important mechanism for central Andean shortening and build three-dimensional geodynamic models to test this hypothesis. Based on the geological constraints on the detailed history of Andean evolution[29] and our model results that produce a similar pattern of Andean shortening, we put forward a model of Andean evolution where the Juan Fernandez Ridge (JFR) acts as a barrier to the northward-migrating trench sediments (Supplementary Fig. 1); the southward migration of JFR since the Eocene (Fig. 1b) creates a southward expanding segment of the sediment-starved trench with enhanced plate coupling and crustal shortening, causing the southward growth of the Andes. This migrating Andean shortening is consistent with the kinematic history of inferred crustal compression. The model results are also consistent with the observed long-strike variation in the magnitude of crustal shortening.

## Results

**Computation of central Andean shortening**. We build numerical models to investigate the role of sediment subduction and JFR migration on Andean shortening (see the "Methods" section, Fig. 2 and Supplementary Figs. 2 and 3). The model starts with a realistic trench configuration of South America (Fig. 2 and Supplementary Fig. 2), which we found to be important for upper plate deformation. Self-consistent free subduction is enabled during the computation, while the initial slab is generated by imposing surface plate motion[22] from 60 to 45 Ma[30] (Supplementary Fig. 2b, c). A lubricating layer with a low viscosity of $10^{19}$ Pa s is defined at the plate interface to decouple the two plates, mimicking the influence of sediments (Supplementary Fig. 3), with a thick (40 km) layer corresponding to weak coupling, and a thin (20 km) layer corresponding to strong coupling (see the "Methods" section). Here, we present three models for comparison with both their final subduction stages (Fig. 2) and the respective time evolution (Supplementary Figs. 4–6). The three models differ in the style of plate coupling, with Model 1 having uniformly weak coupling along the entire trench (Fig. 2d), Model 2 having a stationary strong coupling only in the central region (Fig. 2e), and Model 3 having a southward expanding region of strong coupling (Fig. 2f). The expansion of the strong coupling zone in Model 3 is realized by progressively shifting the

southern edge (at a rate of 0.5°/Ma, Fig. 2f) of the lubricating layer towards the south to mimic the migration of JFR. Models are labeled in the time since starting with 34 Myr roughly corresponding to the present day.

In Model 1, the lubricating layer has a uniform effective thickness of 40 km along the trench (Supplementary Fig. 4). The negative buoyancy of the initial slab drives sustained subduction. We estimate the amount of Andean shortening by comparing the along-strike variation of trench retreat, with a smaller amount of retreat suggesting stronger shortening (see detailed discussion in the "Methods" section). After 32 Myr, most of the shortening occurs between 10°S and 20°S (Fig. 2a), straightening the originally curved trench (green dashed box in Supplementary Fig. 4). Quantitatively, the amount of shortening reaches a maximum value of 150 km between 10°S and 19°S and decreases to zero asymmetrically towards north and south. This shortening is inconsistent with inferences from balanced cross sections[1,2] and paleomagnetism[3] (Fig. 3a) where the amount of shortening reaches a maximum of around 20°S and decreases northward and southward symmetrically. This suggests other factors are controlling the observed pattern of Andean shortening. Additionally, we suggest that an initially curved trench tends to straighten if there are no other forces to balance the local sub-slab pressure gradient imposed by the curvature (Supplementary Fig. 7a).

Model 2 tests the role of enhanced plate coupling in the central Andes due to trench sediment starvation (Fig. 1), where we reduced the thickness of the lubricating layer to 20 km uniformly from 12°S to 33.5°S (Fig. 2e). As a result, the magnitude of the deviatoric stress (second invariant) in the central Andes is significantly increased (Fig. 2b and Supplementary Fig. 5), which causes stronger shortening in the upper plate and thickened crust and higher topography in the central Andes (Fig. 2b). The magnitude of shortening tapers out towards the north and south due to the waning of plate coupling, consistent with the trend of the observed shortening profile (Fig. 3a). However, shortening is over-predicted in the southern central Andes between 20°S to 32°S compared to observation (Fig. 3a). This suggests that enhanced plate coupling could be an important factor in causing the strong shortening in the central Andes, but there exist other mechanisms that reduce the coupling or shortening south of 20°S.

In Model 3, we apply a southward expanding plate coupling zone to mimic the role of the migrating JFR (Fig. 2c and Supplementary Fig. 6). The southern boundary of the plate coupling zone was initially at 18°S, which then gradually migrated to 33.5°S after 31 Myrs at a rate of 0.5°/Ma and remained stationary thereafter (Fig. 2f). Following the southward expansion of the plate coupling zone, the deviatoric stress along the plate interface increases towards the south (Supplementary Figs. 6 and 8). This causes a southward shortening and growth of the Andes (Supplementary Fig. 6). Finally, the predicted cumulative shortening reaches a maximum at 20.5°S, and decreases towards the north and south largely symmetrically, resembling that observed[1–3] (Fig. 3a). This is the best-fit model that reproduces the concave margin and the high topography in the central Andes (Fig. 2c and Supplementary Fig. 9b), with a shortening profile largely consistent with that observed (Fig. 3a). Compared to Model 2, this one predicts less shortening south of 20°S by 50–150 km, due to a later onset of shortening associated with JFR migration. Note that after the passage of JFR, the segment would be experiencing continuous shortening until the present, as sediments remained depleted in the trench and the plate coupling remained strong. The decreasing duration of shortening toward the south provides one explanation for the less modified southern segment.

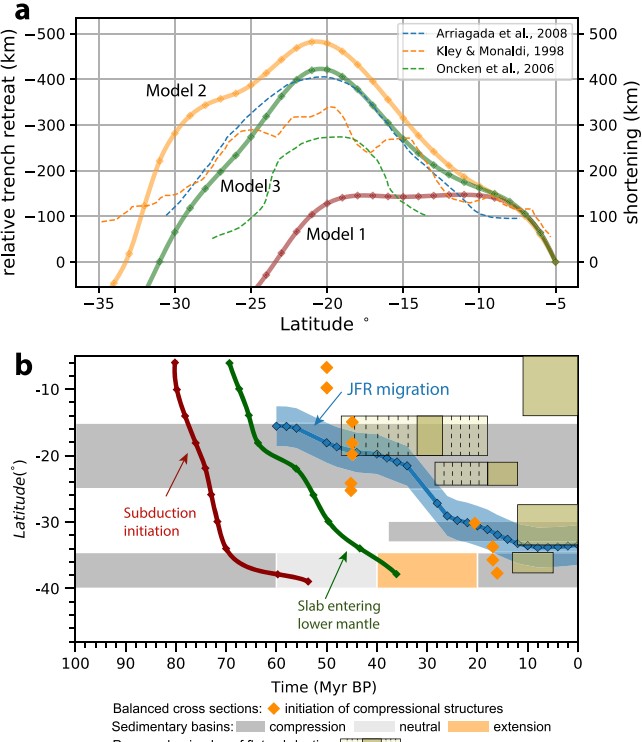

**Fig. 3 Magnitude and onset of shortening in the central Andes. a** Comparison of relative trench retreat as modeled shortening (see the "Methods" section) with observed shortening along strike. **b** Comparison of the JFR migration with the onset of compression. The blue line and diamonds indicate the southward migration of trench–JFR intersection following plate reconstruction[22]. The blue shading represents the region (within 3° from the ridge axis) whose stress regime is directly affected by the collision of JFR with the continent, as revealed by the shoreline indentation of central Chile[27]. The orange diamonds indicate the onset of compressional structures in balanced cross sections[32]. The red and green lines show the timing of Nazca subduction re-initiation and when the slab enters the lower mantle based on seismic tomography[33]. The dark yellow rectangles represent proposed episodes of flat slabs inferred from magmatic distribution and tectonic evolution[45,47], with the dashed area representing the uncertainties associated with the starting and end of the flat slab episodes. The gray, white, and orange rectangles show the compressional, neutral and extensional stress regimes of the Andes constrained by sedimentary records of the foreland and hinterland basins[31,35,37].

**Geologically constrained Andean evolution.** The Andes have experienced multiple stages of compression and extension[31], but the major Andean shortening, where relatively large shortening rates led to the present-day high Andes, is believed to have occurred since the Eocene and is characterized by the activation of the Eocene and post-Eocene compressional structures[9,32]. We compare the migration of JFR to the onset of these compressional structures from balanced cross sections[9,32,33], to illustrate the potential causal relationship (Fig. 3b). Observation shows that the major Andean shortening commenced in the middle Eocene around 15°S, and subsequently expanded southward, with the front of this expansion largely following the migrating JFR. For example, the individual onset of compressional structures along several transects at 20°S, 30°S, and 35°S all match the arrival of JFR (Figs. 3b, 4). The onset of compressional structures along 15°S at 45 Ma postdates the arrival of the JFR (Fig. 3b), but coincides with the Cenozoic cooling of climate[34] (Supplementary Fig. 10a). This is likely because Cenozoic cooling is a necessary

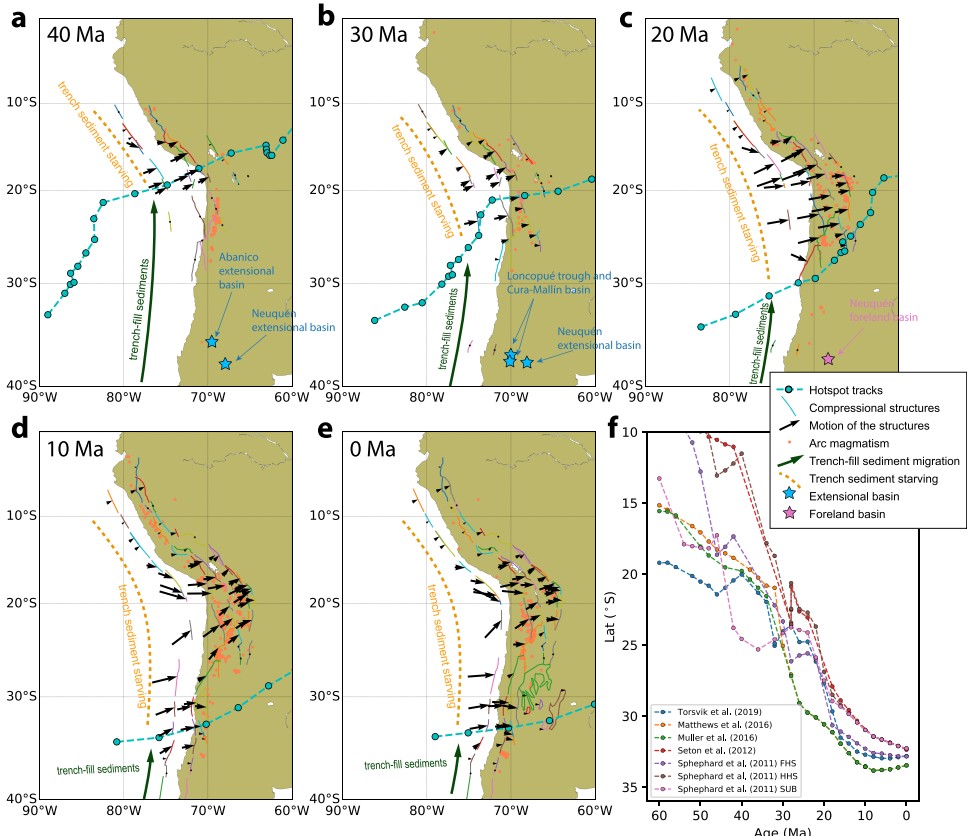

**Fig. 4 Southward propagation of central Andean shortening in response to the migrating JFR.** The kinematics of the Andean tectonic structures[29] and the paleo-position of the JFR based on the reconstruction of Müller et al. [22] are shown for 40 Ma (**a**), 30 Ma (**b**), 20 Ma (**c**), 10 Ma (**d**) and present (**e**), respectively. The history of the sedimentary basins[59] is shown, with extensional basins (blue stars) indicating crustal extension and foreland basins (pink stars) indicating compression. The orange dots represent the arc magmatism from R. Pilger's compilation of Andean Igneous Isotopic Dates[60]. **f** the migration of the intersecting point between the trench and the JFR based on different plate reconstructions[22–26]. FHS fixed hotspot reference frame. HHS hybrid hotspot reference frame. SUB subduction reference frame.

condition to reduce the trench sediments and increase plate coupling by initiating the arid climate in central Andes[13]. However, the transect at about 25°S has an onset age of 45 Ma predating the arrival of JFR at about 30 Ma (Fig. 3b), suggesting other mechanisms could have modified the Andes. On the other hand, foreland and hinterland sedimentary history[31,35] suggest that the central Andes have been in compression since the Late Cretaceous (Fig. 3b). Between 26°S and 28°S, the broken foreland basins could have been partially activated at 55 Ma[36]. At about 30°S, some minor compression could have initiated before the late Eocene as inferred from the slow rate of basin subsidence[37]. We suggest these events could reflect earlier phases of compression before the major Andean shortening. Their corresponding mechanisms will be discussed later.

Compared to the onset of compression which could be complicated by a number of factors, the detailed kinematic history of Andean tectonic structures should provide better constraints on Andean evolution (Fig. 4). The demonstrated kinematic reconstruction is based on shortening estimates and timing constraints from 50 published balanced cross-sections (see the "Methods" section in ref. [29]). We find the paleo-position of the JFR well correlates with these reconstructed motions of the compressional structures (Fig. 4). For example, within the central Andes, before the arrival of the JFR, the compressional structures are usually not activated or their motion is minor; while upon the arrival of the JFR, the compressional structures shift inboard with a fast convergence rate (about 0.5–3 cm/yr), suggesting strong deformation in the upper plate (Fig. 4a–e). The strong

deformation persists long after the JFR has migrated southward away from the region. This observation supports the southward expanding plate coupling zone in the wake of JFR as a key driving mechanism for Andean shortening. We speculate that the collision of JFR with the overriding plate could have facilitated the subsequent long-term shortening by creating the first phase of structural damage to the overriding plate.

## Discussion
According to our model, the southward expansion of plate coupling is controlled by the along-strike variation of sediment influx, trench axial sediment transport, and the topographic barrier from the migrating JFR (Fig. 4a–e). Climate forces are critical in controlling sediment influx[13,15]. The present Andes is characterized by arid climate resulting from the subtropical atmospheric subsidence, the temperature inversion due to the cold Humboldt Current (Supplementary Fig. 11), and a rain-shadow of the high Andes associated with the westward trade wind[38–40]. These climate factors lead to reduced sediment supply to the trench from surface erosion and enhanced plate coupling that can cause strong deformation in the central Andes and supports the present-day high topography (Fig. 2). As proposed earlier[13,15], the initiation of the arid climate system is likely due to the Cenozoic cooling[34] (Supplementary Fig. 10a). This is consistent with the 45 Ma onset age of shortening in southern Peru which postdates the arrival of JFR (Fig. 3b) but coincides with the onset of cooling[34].

Largely neglected in earlier studies is the effect of the climate forces in the southern Andes. Our model suggests the relatively rich sediment supply in the southern Andes plays an important role in controlling the pattern of plate coupling. These sediments feed to the trench south of JFR along the arid central Andes through trench axial channels[17,18,21] (Fig. 4a–e) and decouple the two plates, which could result in a later onset and a reduced magnitude of shortening south of 20°S (Fig. 3a). The rich sediment supply is likely caused by high precipitation (Supplementary Fig. 11) due to the prevailing westerlies that carry the moisture from the Pacific[41], enhanced tectonic erosion following tectonic uplift[42], and a glacial "buzzsaw" effect at midlatitude[41,43].

The relatively rich trench sediments in the southern Andes could also be associated with tectonic factors. Variation of plate convergence along the Andes[22] (Supplementary Fig. 10b) could have influenced the consumption rate of trench sediments due to subduction. Before 30 Ma, the convergence rate in the southern Andes was less than half of that of the central Andes (Supplementary Fig. 10b). Therefore, a slower consumption rate of trench sediments was expected in the southern Andes, leading to the preservation of thick trench-fill sediments. We suggest a sediment thickness of ≥1 km is likely sufficient to decouple the two plates, as is the case for the Andes north of 10°S.

Tectonically, multiple mechanisms have been proposed to account for the central Andean shortening, including re-initiation of subduction[33], changes in slab dip angle[44,45], variation of overriding plate velocity[31], changes in seafloor age[5], and slab-lower mantle interaction[9,10,12]. Multiple geological events could be linked to these mechanisms. For example, the fast westward advance of South America has been proposed to account for the Late Cretaceous onset of compression for the central and southern Andes as revealed by sedimentary records in backarc basins[31] (Fig. 3b). This compression significantly predated the major central Andean shortening. The early onset of compressional structures at 45 Ma along the transect at 25°S (Fig. 3b) could be attributed to slab-lower mantle interaction. Similarly, the Eocene compression at about 30°S inferred from basin subsidence[37] (Fig. 3b) could also be explained by this mechanism. In addition, a recent study[36] proposed this mechanism to be responsible for the formation of the broken foreland between 26°S and 28°S from 55 to 30 Ma. While we agree this mechanism could modify the Andean shortening, we speculate it only caused mild shortening and thus is not the main contributor for the major Andean shortening. This is because strong shortening was absent from the kinematic reconstruction of the compressional structures at these latitudes before and during the Eocene[29] (Fig. 4). The inferred basin subsidence around 30°S has a slow rate in the Eocene compared to the later fast subsidence since the late Oligocene–early Miocene[37] that was interpreted to be the onset of the most recent phase of major Andean shortening at this latitude[46].

Of particular interest is the combination of two mechanisms, the slab-lower mantle interaction[9], and the southward propagating subduction re-initiation[33]. In this scenario, the slab enters the lower mantle first in Peru and then propagates southward, causing the southward migration of shortening along the Andes (Fig. 3b). We challenge this hypothesis with two arguments. First, a nonuniform 10–30 Myrs time delay has been inferred along the Andes between the timing of slab penetration into lower mantle and the onset of compressional structures by reconstructing Nazca subduction using seismic tomography and constrains from magmatism[33] (Fig. 3b). Second, according to this model, a larger amount of shortening is expected in the north than the south due to the longer duration of slab-lower mantle interaction—contradicting the observed northward tapering of shortening from northern Chile to northern Peru (Fig. 3a). We suggest southward

expanding plate coupling due to sediment migration and subduction is a more likely mechanism, given the inverse correlation between sediment thickness and mountain height (Fig. 1c), the match between the strong-deformation front in the central Andes and the migration of JFR (Fig. 4), as well as the fit of the model that includes southward expanding plate coupling to the observed magnitude of shortening (Fig. 3a).

Flat slab subduction is another widely adopted mechanism for Andean shortening[31,45,47]. Multiple episodes of flat slabs have been proposed based on the magmatic distribution and tectonic structures of the Andes[45] (Fig. 3b). These episodes of flat slabs have been proposed to be responsible for the formation of the broken forelands[39,45] and the widening of the Andes[47] (Fig. 3b). For example, in the Sierras Pampeanas, the thick-skinned deformation since the mid-Miocene coincides with the onset of Pampeanas flat slab, with the latter widely suggested as the cause of the former[39,45,48]. Therefore, a flat slab may be an important factor that contributes to the formation of the Andes. In fact, the flat slab and the model proposed by this study are not mutually exclusive. Subduction of the JFR has been proposed to be one mechanism for the formation of flat slab[30,44], However, the proposed flat slabs in the central Andes from 15°S to 25°S do not continue to the present day (Fig. 3b). Therefore, the compressional force that maintains the long-term shortening is more likely related to enhanced plate coupling due to trench-sediment depletion. On the other hand, the broken foreland, as characterized by thick-skinned deformation, cannot be fully explained by flat slabs[47] or the model proposed by this study, because some deformation was as early as 55 Ma[36], predating flat-slab subduction and the arrival of JFR. This deformation has been proposed to be related to pre-existing structures in the basement[39,49] and facilitated by tectonic compression that could be induced by slab–lower mantle interaction[36]. We suggest this deformation does not necessarily represent the start of the major Andean shortening. We emphasize the formation of the broken foreland represents extra complexities for the evolution of the Andes, since thick-skinned deformation may result in a reduced shortening rate compared to the preceding thin-skinned deformation, as is the case in the Sierras Pampeanas[45]. It may also be more difficult to measure the shortening for thick-skinned deformation due to the lack of stratigraphic markers. Therefore, it introduces extra uncertainty for the distinguishing of model 2 and model 3 using the amount of shortening. We will rely on future studies for a better understanding of these two deformation styles.

In summary, we define two phases of compression during central Andean evolution, with the first phase corresponding to the Paleocene–Eocene compression as recorded by sedimentary basin processes[31,37,50], and the second phase to be the major central Andean shortening that initiated between Eocene and early Miocene depending on the latitude (Fig. 5). In contrast to the broad-scale major central Andean shortening that affected regions from the cordillera to the Subandean, the Paleocene–Eocene compression affected a much-focused area mostly in the Precordillera and the Western Cordillera[31,51] (Fig. 5a), possibly with reactivation of some pre-existing structures in the foreland of the southern central Andes[36]. We suggest this earlier phase of compression that might be traced back to the Late Cretaceous[31], is likely caused by the gradual shallowing of the slab with subduction duration[52], rapid convergence between subducting and overriding plates[31], or slab penetration[9] and interaction with the lower mantle[11,12]. This compression caused low-magnitude crustal shortening (Fig. 5a), widely observed in compressive orogens. Note the southern Andes (south of 35°) experienced a synchronous neutral-to-extensional regime at that time (Fig. 3b). The compression that caused low-magnitude shortening[53] was triggered later at about 20 Ma (Fig. 3b). This

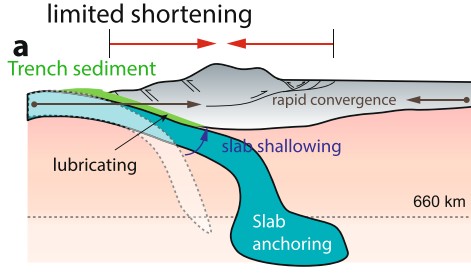

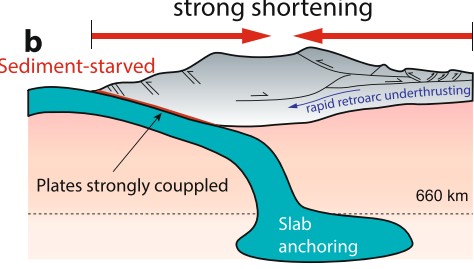

**Fig. 5 Schematic showing two phases of compression during Andean evolution. a** The first phase of compression is accompanied by mild shortening due to gradual shallowing of the slab with increasing subduction duration, slab–lower mantle interaction through the anchoring effect, and the rapid plate convergence. This occurred during Late Cretaceous to early Eocene when thick sediments were subducted. **b** The second phase of compression is accompanied by strong shortening due to enhanced plate coupling, which causes the main phase of Andean mountain building. This has been operating since Eocene following the arrival of JFR that deprived trench sediments in the north. The thick green line along the plate interface in **a** indicates trench-fill and subducted sediments. The red line along the plate interface in **b** indicates the sediment-depleted trench and subduction channel, suggesting a strong coupling between the two plates.

may be due to the higher sediment influx and lower sediment consumption in the southern Andes as discussed earlier, or because the southern Andes is more approximal to the edge of the subduction zone[11], both of which tend to mitigate backarc compression.

We suggest the latter phase of compression in the central Andes is mainly due to lack of sediment subduction. Since the Eocene, due to global cooling[34] and the resulting arid climate in the central Andes, sediment supply to the trench was greatly reduced. This caused increased plate coupling and initiated the atypical extensive large-magnitude shortening that was accommodated by rapid retroarc underthrusting[54] and deformation within the crust with variable strength[55] (Fig. 5b). This compression, with flat slabs episodically modifying the Andes (Fig. 3b), could account for the main phase of Andean mountain building. We propose the plate coupling zone was first initiated in the north and then expanded towards the south, controlled by sediment influx and transport modulated by the southward migrating JFR. This explains the diachronous strong deformation since the middle Eocene as represented by the kinematics of the compressional structures (Fig. 4). We suggest trench-sediment transport and subduction are important processes that have a profound effect on the dynamics of plate boundaries and orogenesis. This model represents an example of cross-scale interaction in geology: a narrow "dam" to sediment transport in the trench could exert a strong influence on the formation of Andes.

## Methods

**Numerical model.** We use the open-source finite-element code CitcomS[56,57] for the computations. Assuming an incompressible mantle that satisfies the Boussinesq approximation, we solve for thermal–chemical convection governed by conservation of mass, momentum, and energy. The model setup and choice of parameter values are based on our earlier data-assimilation models[30] that fit both upper mantle Benioff zone shapes and lower mantle structure from seismic tomography[30]. Some changes are made for this study, including a viscosity law with both diffusion and dislocation creep, a free-slip boundary condition at the top surface that allows for free subduction, and a higher Rayleigh number equal to $3.38 \times 10^8$ which corresponds to a mantle temperature of 1300 °C and a reference viscosity of $10^{21}$ Pa s (Supplementary Table 1).

To reiterate, a regional domain is used for all cases which cover the whole South America and the surrounding area from 30 to 120°W, 30°N–70°S, and radially from the core–mantle boundary to the surface. The model domain is discretized into $768 \times 512 \times 64$ elements in longitude, latitude, and radius, respectively, with spatially variable mesh sizes and a resolution up to 10 km at the subduction zone.

The viscosity relation used is

$$\eta(T,r) = A(r)\eta_c \dot{\epsilon}_{\mathrm{II}}^{\frac{1-n}{n}} \exp\left(\frac{E_a}{T + T_{\mathrm{off}}} - \frac{E_a}{1 + T_{\mathrm{off}}}\right) \qquad (1)$$

where $\eta$ is non-dimensional viscosity, $T$ is temperature, $r$ is the radius, $A$ is the viscosity pre-factor, $\eta_c$ is the intrinsic composition-dependent pre-factor, $E_a$ is the activate energy with a non-dimensional value of 18, $T_{\mathrm{off}}$ is the temperature offset that has a value of 0.1 in the upper mantle and 0.6 in the lower mantle, $\dot{\epsilon}_{\mathrm{II}}$ is the second invariant of the strain rate tensor, $n$ is the stress exponent which equals 1.0 for diffusion creep and 3.5 for dislocation creep (Supplementary Table 2). The viscosity pre-factor for diffusion creep equals 0.25 in the upper mantle and transitions smoothly to 20 in the lower mantle. The dislocation creep has a viscosity pre-factor of $5 \times 10^{-10}$, causing up to 1–2 orders of magnitude viscosity reduction in the upper mantle relative to the diffusion creep.

The final effective viscosity is controlled by the composite rheology of diffusion and dislocation creep and the pseudo-plasticity

$$\eta_{\mathrm{eff}} = \min\left(\frac{\sigma_y}{\dot{\epsilon}_{\mathrm{II}}}, \frac{\eta_{\mathrm{diff}}\eta_{\mathrm{disc}}}{\eta_{\mathrm{diff}} + \eta_{\mathrm{disc}}}\right) \qquad (2)$$

where $\sigma_y$ is the yield stress which equals 200 MPa throughout the mantle.

We use tracers (Lagrangian particles) to define the composition field. The geometry and properties of different compositions are shown in Supplementary Fig. 3. A layered structure with uniform crust and lithosphere for both oceanic and continental plates is used. The continental lithosphere is further decomposed into the cratonic and non-cratonic regions. The shape of the craton is shown in Fig. 2, which resembles the present Amazonian Craton. Due to the limitation of the resolution, we set the thickness of the oceanic crust to be 25 km and the density to be 3200 kg/m³. The total buoyancy is equivalent to the typical oceanic crust of the Earth which has a density of 2900 kg/m³ and a thickness of 8 km. We set the initial continental crust to be 40 km thick with a density of 2800 kg/m³. The density of the continental mantle lithosphere is compositionally neutral with a thickness of 80 km for the non-cratonic region and 110 km for the cratonic region. This leads to a total thickness of 120 km for the non-cratonic lithosphere and 150 km for the cratonic lithosphere. The viscosity of the lithosphere is set to be 100 times the upper mantle pre-factor for the non-cratonic region and 2000 times the upper mantle pre-factor for the craton. A conductive thermal profile is initially set for the continent, of which the non-dimensional temperature decreases from 1.0 at the base of the continent to 0.0 at the surface. For the oceanic plate, we set the initial age to be 50 Myrs, and use half-space cooling to define the thermal structure.

For plate coupling, we define a lubricating layer at the plate interface (Supplementary Fig. 3), whose effect is similar to the subduction channel in earlier studies on plate coupling[14] and to the weak zone in many other geodynamic models[9,12]. First, we monitored the composition of the elements vertically from the surface downward into the mantle. The first element that contains the tracers of the oceanic crust is defined as the element of the plate interface. By searching all elements near the subduction zone, a curved plate interface is constructed that conforms to the top surface of the slab. The non-dimensional viscosity within a certain distance of the plate interface is reduced to 0.01, to mimic the lubricating effect. Viscosity smoothing is applied to suppress large gradients. For a coupled case, the effective width of the lubricating layer is 20 km, while for a relatively uncoupled case, it is 40 km. These numbers are likely larger than the actual thickness of the subduction channel. They represent a compromise given the finite model resolution. By controlling the latitudinal ranges of the thin and thick lubricating layers, we test the pattern of plate deformation in response to the variation in plate coupling.

To generate the initial condition, we impose the surface plate motion from 60 to 45 Ma, which essentially generates several hundred kilometers of initial slabs at the subduction zone (Supplementary Fig. 2). Then we restart the model with a free-slip top boundary condition to initiate the free subduction. The model runs forward for 30–40 Myrs and the resultant plate deformation is compared to geological observations.

**Estimating shortening**. It is difficult to determine the absolute value of shortening in 3D geodynamic models because of the existence of subduction erosion at the leading edge of the overriding plate and distributed deformation within the plate. Following the method in ref. [11], we determine the relative shortening by comparing the variation of trench migration along strike. We first determine the trench location when the model starts by tracking the compositional boundary between oceanic and continental plates. Subsequently, we determine the trench location when the model ends using the same method. The two trench locations are compared to compute the lateral migration of individual points along the trench. We use the trench migration at 5°S as a reference. For any trench segment, a larger trench retreat than that at 5°S means extension, while a smaller trench retreat means shortening. Consequently, we are able to determine the shortening curves as presented in Fig. 3a.

This only provides the amount of shortening/extension in a relative sense. However, geological observations[1,29,32] suggest a minor shortening at 5°S. This essentially puts the observed and geodynamically predicted shortening in the same reference frame so that they can be compared.

**Reconstruction of JFR**. The southward migration of JFR has been proposed by earlier studies[27,28]. With recent advances in plate reconstruction models, the path of JFR migration could be updated and more carefully examined. We assume a fixed Juan Fernandez hotspot at 79°W, 34°S, and use multiple plate reconstruction models[22–26] to reconstruct the paleo-position of JFR. We first compute the present trajectory (partly subducted) of the JFR using the Euler poles of the Nazca Plate and the location of the Juan Fernandez hotspot. Then the whole trajectory is reconstructed backward in time by calculating the motion of the Nazca Plate relative to the South American plate. The intersection between the JFR and the pre-Andean trench[3] is then computed at different ages, giving JFR migration curves in Fig. 4f.

We find all plate reconstructions produce a southward migrating JFR, suggesting it is a robust feature. Different models predict slightly different locations for the intersection point at particular ages. The difference is minor among the recent models after 2016 that have incorporated updates and revisions on the absolute reference frame[22,24] and plate circuits[25]. We refer to these models to reconstruct the Andean evolution. Fig. 4a–e only shows the reconstructed JFR based on ref. [22], but we emphasize the other two models[24,25] yield similar JFR migration.

## Data availability

The data supporting the findings of this study are available within the paper. The Source Data of the figures are provided as a supplementary file. Source data are provided with this paper.

## Code availability

The computational code CitcomS is available at the Computational Infrastructure for Geodynamics (CIG), https://geodynamics.org/. The python library PyGPlates that provides the functionality to reconstruct the paleo-tectonic features can be found at https://www.gplates.org/.

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

## Acknowledgements

We used the open-source software PyGPlates to reconstruct the paleo-structures. L.L. and J.H. were supported by the National Science Foundation through EAR-1565640 and EAR-1554554. M.G. and J.H. were supported by the National Science Foundation through EAR-1645775. J.H. was also partially supported by the National Science Foundation of China through project 42174106. Computations were carried out on the NSF-supported Frontera supercomputers at the Texas Advanced Computer Center and were also supported by Center for Computational Science and Engineering of Southern University of Science and Technology.

## Author contributions

J.H. conceived the study and performed the numerical experiments. L.L. and M.G. provided valuable advices throughout the study. All the authors participated in manuscript preparation.

## Competing interests

The authors declare no competing interests.

## Additional information

**Peer Review Information** *Nature Communications* thanks Sebastian Zapata for their contribution to the peer review of this work. Peer reviewer reports are available.

