## [Peer Review File · Nature Communications]

Editorial Note: This manuscript has been previously reviewed at another journal that is not operating a transparent peer review scheme. This document only contains reviewer comments and rebuttal letters for versions considered at Nature Communications .

REVIEWER COMMENTS

Reviewer #1 (Remarks to the Author):

After two rounds of reviews, I have been asked to present my position on the comments from previous reviewers. Despite I will not make detailed comments in the text, I also present few additional comments from my review.

General comment

I consider this is a really interesting contribution that needs to be presented to the community, this manuscript presents 2 points that will impact our understanding of Andean tectonics.

1 The absence of trench sediments can promote extensive upper plate compression by increasing plate coupling.

2 The southward migration of the JFR can explain some of the along-strike variation in the deformation patterns observed in the Central Andes.

I consider that observations made for previous reviewers do not compromise the main conclusions of this manuscript. However, some points can be better addressed in the manuscript. This manuscript presents sediment supply as the main or only precursor of the later phases of deformation in the Andes, but the role of other tectonic processes it is not discussed enough. These issues can be solved by improving the comparison with other tectonic processes and clarifying some statements. More detailed comments are below.

Comments on previous reviews

Regarding the comments from reviewer 1, which suggest that the crustal thickness in the model is inconsistent with the observations. I consider the author's answers are satisfactory since rheological simplification is required for the model and all variables can not be taken into account. Concerning the comments on the slab angle, this is indeed an important feature traditionally invoke to explain retroarc compression [1]–[3], I am not sure if it's ok to say that it played a minor role, as the authors suggested in their answer. But I do agree in two critical points, the model is designed to assess if the trench starvation can produce the required compression and thus explain the Oligocene to present along-strike deformation patterns; despite flat-slab may have influenced across-strike deformation patterns, it's not enough to explain the along-strike deformation patterns [4]. In general, I agree with the authors that model simplifications are required and the model setup is oriented towards a research question that does not involve the subduction angle. Of course, many questions remain open but they will have different setups that is beyond the scope of this contribution.

Reviewer 1 suggests that the ridge is not the only reason to explain the lack of sediments in the trench and suggests that the rain shadow and the dry climate in the western side of the Andes played a key role. These two processes are not mutually exclusive. I agree with the evident first-order control of the JFR discussed and presented by the authors, additionally, the authors explained in the manuscript that minor contribution of continental sediments is necessary to maintain a high plate coupling, thus, I consider that additional modifications are not necessary on this topic. Finally, reviewer one makes a good point saying that the manuscript suggests that the major compression correlates with the position of the JFR, but it doesn't necessarily mean that trench starvation is the mechanism, why not flat slab? which has been widely proposed as a precursor of compression in the Central Andes [1], [2], [5]. Despite, I consider that the authors present a reasonable answer, this needs to be better presented in the manuscript, I recommend the authors include boxes in Fig 3b to indicate periods of flat-slab subduction in the study area; since the authors are already comparing against slab anchoring, they should also include flat-slab.

I do agree with most of the comments from reviewer 3 and I consider they can be better addressed, especially comments 1 and 3.

In comment 1 the reviewer 3 indicates that the fit is not perfect and the authors agree to it. However, I consider additional discussion and comparison are required. In comment 3 reviewer 3 points to the flaws of shortening metrics and the relevance of deformational styles to understand deformation patterns, I think that the answer of the authors in this topic is not satisfactory, as reviewer 3 also suggested. Below I left some specific comments and recommendations to address these issues.

Flat-slab can explain local mismatches between the modeled and observed data, this is an additional reason to include flat-slab patterns in Fig 3b, as I already suggested. The match Between the position of the JFR and the observed data is not perfect. This suggests that there are more processes involve and the JFR

migration is only partially responsible for the Andean compression. The authors mentioned plausible explanations for some mismatches but did not discuss that the model works poorly in the southern broken forelands. In the Sierras Pampeanas and Santa Barbara broken forelands between 22°S to 30°S, the data that the authors presented is 10 Ma older than the arrival of the JFR or there is no data. The lack of shortening data is likely related to the deformational style since in thick-skin deformation the assessment of shortening can be challenging due to the lack of stratigraphic markers. But the lack of shortening data does mean that there is no data about the time of compression. Available thermochronological and field data suggest deformation at ~55 Ma and after 15 Ma [4], [6]–[8], none of these phases match the arrival of the JFR at 30 Ma.

This is critical to this research for two reasons:

- a) The authors are using the amount of shortening in this region (25S to 33S) to rule out model two (which do not require a southward migration of the JFR), so the estimation of shortening in this region is critical to distinguish between model two and three. So basically, a poorly constrained amount of shortening in this region is used to favor the southward propagation model. Models 2 and 3 are similar, then model 2 needs to be more discussed.
- b) The model doesn't seem to work in the broken foreland region, which is interesting and needs to be discussed.

Finally, I have two additional comments.

The authors have separated Andean compression into two phases, and then used the terms "Major Andean compression" (L139) and "First compressional structure" in Fig 3. Since the time of major Andean compression (not the onset) is used to validate the model. I consider the authors should define clearly how they defined this. It is the first post-Eocene compressional structure??? Or there is a shortening rate threshold after which the authors defined the onset of "major Andean shortening"?? Or is the onset of retroarc deformation?? I know that in several segments of the manuscript the authors explained or mentioned this issue. But I think a clear definition or threshold needs to be established.

The presented model explains some of the Oligocene to present Andean deformation, as the authors also explained and Andean growth has been occurring before the arrival of the JFR, in minor proportions or with other deformational styles... but deformation and 'growth' is not fully related to the JFR. Moreover, the authors also acknowledge that in some segments flat-slab controlled deformation. Please consider changing some segments manuscript where this model is presented as the only one responsible for Andean topography (you are discussing topographic growth), widening, and deformation, which is clearly not the case (e.g. L21, L44, L177, L264.. and others). Acknowledging the complexities of the tectonic processes that were responsible for Andean growth will not diminish the impact of this manuscript.

I don't consider that these issues should prevent the manuscript from being published, but I do consider that additional discussion and clarifications are required.

References

- [1] B. K. Horton, "Tectonic Regimes of the Central and Southern Andes: Responses to Variations in Plate Coupling During Subduction," *Tectonics*, vol. 37, no. 2, pp. 402–429, 2018.
- [2] J. Martinod, M. Gérard, L. Husson, and V. Regard, "Widening of the Andes: An interplay between subduction dynamics and crustal wedge tectonics," *Earth-Science Rev.*, no. March 2020.
- [3] V. A. Ramos and A. Folguera, "Andean flat-slab subduction through time," *Geol. Soc. London, Spec. Publ.*, vol. 327, no. 1, pp. 31–54, 2009.
- [4] S. Zapata, E. R. Sobel, C. Del Papa, and J. Glodny, "Upper plate controls on the formation of broken foreland basins in the Andean retro-arc between 26 and 28° S: from Cretaceous rifting to Paleogene and Miocene broken foreland basins," *Geochemistry, Geophys. Geosystems*, vol. 21, no. 7, p. e2019GC008876, 2020.
- [5] V. A. Ramos, E. O. Cristallini, and D. J. Pe, "The Pampean flat-slab of the Central Andes," *J. South Am. Earth Sci.*, vol. 15, no. 1, pp. 6–8, 2002.
- [6] E. R. Sobel, G. E. Hilley, and M. R. Strecker, "Formation of internally drained contractional basins by aridity-limited bedrock incision," *J. Geophys. Res. Solid Earth*, vol. 108, no. B7, 2003.
- [7] P. Payrola et al., "Episodic out-of-sequence deformation promoted by Cenozoic fault reactivation in NW Argentina," *Tectonophysics*, vol. 776, p. 228276, 2020.
- [8] A. L. S. Goddard et al., "Reconstructing the thermal and exhumation history of the Sierras Pampeanas through low-temperature thermochronology: A case study from the Sierra de Velasco," no. Xx, pp. 1–17, 2018.

Reviewer #1 (Remarks to the Author):

After two rounds of reviews, I have been asked to present my position on the comments from previous reviewers. Despite I will not make detailed comments in the text, I also present few additional comments from my review.

General comment

I consider this is a really interesting contribution that needs to be presented to the community, this manuscript presents 2 points that will impact our understanding of Andean tectonics.

1 The absence of trench sediments can promote extensive upper plate compression by increasing plate coupling.

2 The southward migration of the JFR can explain some of the along-strike variation in the deformation patterns observed in the Central Andes.

I consider that observations made for previous reviewers do not compromise the main conclusions of this manuscript. However, some points can be better addressed in the manuscript. This manuscript presents sediment supply as the main or only precursor of the later phases of deformation in the Andes, but the role of other tectonic processes it is not discussed enough. These issues can be solved by improving the comparison with other tectonic processes and clarifying some statements. More detailed comments are below.

We appreciate the reviewer's support. The concerns raised by the reviewer have been addressed in this revision. Please see our detailed responses below.

Comments on previous reviews

Regarding the comments from reviewer 1, which suggest that the crustal thickness in the model is inconsistent with the observations. I consider the author's answers are satisfactory since rheological simplification is required for the model and all variables can not be taken into account. Concerning the comments on the slab angle, this is indeed an important feature traditionally invoke to explain retroarc compression [1]–[3], I am not sure if it's ok to say that it played a minor role, as the authors suggested in their answer. But I do agree in two critical points, the model is designed to assess if the trench starvation can produce the required compression a thus explain the Oligocene to present along-strike deformation patterns; despite flat-slab may have influenced across-strike deformation patterns, it's not enough to explain the along-strike deformation patterns [4]. In general, I agree with the authors that model simplifications are required and the model setup is oriented towards a research question that does not involve the subduction angle. Of course, many questions remain open but they will have different setups that is beyond the scope of this contribution.

Reviewer 1 suggests that the ridge is not the only reason to explain the lack of sediments in the trench and suggests and that the rain shadow and the dry climate in the western side of the Andes played a key role. These two processes are not mutually exclusive. I agree with the evident first-order control of the JFR discussed and presented by the authors, additionally, the authors explained in the manuscript that minor contribution of continental sediments is necessary to maintain a high plate coupling, thus, I consider that additional modifications are not necessary on this topic.

We appreciate the reviewer's support to our previous response to reviewer 1, including the rheological simplifications and the first-order control of JFR on sediment distribution.

Finally, reviewer one makes a good point saying that the manuscript suggests that the major compression correlates with the position of the JFR, but it doesn't necessarily mean that trench starvation is the mechanism, why not flat slab? which has been widely proposed as a precursor of compression in the Central Andes [1], [2], [5]. Despite, I consider that the authors present a reasonable answer, this needs to be better presented in the manuscript, I recommend the authors include boxes in Fig 3b to indicate periods of flat-slab subduction in the study area; since the authors are already comparing against slab anchoring, they should also include flat-slab.

This is a good suggestion. We have updated the manuscript by including boxes in Fig. 3b to indicate possible periods of flat slabs. We have also discussed the flat slab hypothesis. Please see our response below.

I do agree with most of the comments from reviewer 3 and I consider they can be better addressed, especially comments 1 and 3. In comment 1 the reviewer 3 indicates that the fit is not perfect and the authors agree to it. However, I consider additional discussion and comparison are required. In comment 3 reviewer 3 points to the flaws of shortening metrics and the relevance of deformational styles to understand deformation patterns, I think that the answer of the authors in this topic is not satisfactory, as reviewer 3 also suggested.

We thank the reviewer for the constructive suggestions. Please see our responses below.

Below I left **some specific comments and recommendations** to address these issues.

Flat-slab can explain local mismatches between the modeled and observed data, this is an additional reason to include flat-slab patterns in Fig 3b, as I already suggested.

We agree with the reviewer. We have included periods of flat slabs in Fig. 3b and used one paragraph to discuss its effect (Line 260-274). We agree that flat slabs have participated in the formation of the Andes. In the Sierras Pampeanas, the thick-skinned deformation since the mid-Miocene coincides with the onset of Pampeanas flat slab, suggesting a causal relationship (Ramos and Folguera, 2009). It has also been suggested that flat slabs can widen the Andes by initiating compression far from the trench (Martinod et al., 2020). Although we agree these processes could have occurred during the formation of the Andes, we proposed the long-lasting force that has caused continued shortening even when the flat slab re-steepens (e.g. between 15-25°S) (which tends to cause extension during the re-steepening stage) comes from the strong plate coupling due to trench-sediment starvation.

The match Between the position of the JFR and the observed data is not perfect. This suggests that there are more processes involve and the JFR migration is only partially responsible for the Andean compression. The authors mentioned plausible explanations for some mismatches but did discuss that the model works poorly in the southern broken forelands. In the Sierras Pampeanas and Santa Barbara broken forelands between 22°S to 30°S, the data that the authors presented is 10 Ma older than the arrival of the JFR or there is no data. The lack of shortening data is /likely related to the deformational style since in thick-skin deformation the assessment of shortening can be challenging due to the lack of stratigraphic markers. But the lack of shortening data does mean that there is no data about the time of compression. Available thermochronological and field data suggest deformation at ~55 Ma and after 15 Ma Ma [4], [6]–[8], none of these phases match the arrival of the JFR at 30 Ma.

This is critical to this research for two reasons:

- a) The authors are using the amount of shortening in this region (25S to 33S) to rule out model two (which do not require a southward migration of the JFR), so the estimation of shortening in this region is critical to distinguish between model two and three. So basically, a poorly constrained amount of shortening in this region is used to favor the southward propagation model. Models 2 and 3 are similar, then model 2 needs to be more discussed.
- b) The model doesn't seem to work in the broken foreland region, which is interesting and needs to be discussed.

This is a good suggestion. Indeed, there could be other processes at play, which could include processes from both the crust and mantle that are not considered in the models.

As mentioned by the reviewer, the thick-skinned deformation at 55 Ma could not be explained by either strong plate coupling due to sediment starvation or flat slab subduction. It has been suggested they were related to the reactivation of pre-existing basement-involved structures (Payrola et al., 2020; Zapata et al., 2020; Goddard et al., 2018). In this case, it may not require a strong compression to reactivate these structures. Slab-lower mantle interaction could be one feasible mechanism (Zapata et al., 2020); this has been discussed in Line 272-277.

We agree the thick-skinned deformation in the broken forelands could introduce additional uncertainty when using the amount of shortening to distinguish model 2 and 3. We point out that model 2 and 3 can also be distinguished by the kinematic reconstruction of compressional structures, where we see large deformation (with relatively large arrows) propagated southward from 20°S to 35°S, coinciding with the migration of JFR (Fig. 4). In addition, although data for the onset age of shortening is lacking between 26-29°S, at ~30°S the onset age of major Andean shortening is about 20 Ma from balanced cross sections (Fig. 3b). This is consistent with basin subsidence that revealed a slow subsidence rate in Eocene-Oligocene and a much faster subsidence rate since Miocene (Fosdick et al., 2017). Therefore, the onset age at ~30°S also support model 3.

Following the reviewer suggestion, we have now acknowledged the drawback of using the amount of shortening (Line 277-283).

Finally, I have two additional comments.

The authors have separated Andean compression into two phases, and then used the terms “Major Andean compression” (L139) and “First compressional structure” in Fig 3. Since the time of major Andean compression (not the onset) is used to validate the model. I consider the authors should define clearly how they defined this. It is the first post-Eocene compressional structure??? Or there is a shortening rate threshold after which the authors defined the onset of “major Andean shortening”?? Or is the onset of retroarc deformation?? I know that in several segments of the manuscript the authors explained or mentioned this issue. But I think a clear definition or threshold needs to be established.

This is a good suggestion.

Ideally, we should use a shortening rate threshold to define “major Andean shortening”. However, this threshold is practically unavailable because the shortening rate is not well constrained, and the tectonic conditions vary greatly along different transects, which makes it inappropriate to use a certain value for the threshold.

To make the logic smoother, we simply use the activation of compressional structures from balanced cross sections to define the major Andean shortening. In other words, the onset of these compressional structures means the onset of major Andean shortening. We believe this is the best observable we could use to define major Andean shortening. We now clarify this in Line 145-148.

However, we emphasize this definition could be flawed. For example, at 25°S the onset of compression is at 45 Ma (Fig. 3b). This is consistent with the kinematic reconstruction of compressional structures that shows the activation of faults before 40 Ma, but these faults barely have any motion, suggesting strong shortening had not occurred at 40 Ma (Fig. 4a).

The first phase of compression is simply defined to be the compression before the major Andean shortening that could be caused by other mechanisms other than the strong plate coupling due to sediment starvation. We have now revised the manuscript, avoiding the term “first compression” until in the summarizing paragraphs (the last two paragraphs) where we compare and contrast the two phases of compression and make it clear that we believe the “major Andean shortening” is the later phase of shortening that initiated between Eocene and early Miocene depending on the latitude, affected a broad region from the cordillera to the Subandean, and had a large magnitude (shortening rate).

This way, we believe we have introduced this concept naturally.

The presented model explains some of the Oligocene to present Andean deformation, as the authors also explained and Andean growth has been occurring before the arrival of the JFR, in minor proportions or with other deformational styles... but deformation and 'growth' is not fully related to the JFR. Moreover, the authors also acknowledges that in some segments flat-slab controlled deformation. Please consider changing some segments manuscript where this model is presented as the only one responsible for Andean topography (you are discussing topographic growth), widening, and deformation, which is clearly not the case (e.g. L21, L44, L177, L264.. and others). Acknowledging the complexities of the tectonic processes that were responsible for Andean growth will not diminish the impact of this manuscript.

Thanks for pointing this out. We have changed these discussions by also acknowledging mechanisms including flat slabs on the evolution of Andes. We have also toned down the discussion of the dominant role of our proposed model throughout the text.

I don't consider that these issues should prevent the manuscript from being published, but I do consider that additional discussion and clarifications are required.

We thank the reviewer for the support on the publication of this manuscript. We hope the newly added discussions and clarifications are satisfactory to the reviewer.

- [1] B. K. Horton, "Tectonic Regimes of the Central and Southern Andes: Responses to Variations in Plate Coupling During Subduction," *Tectonics*, vol. 37, no. 2, pp. 402–429, 2018.
- [2] J. Martinod, M. G rault, L. Husson, and V. Regard, "Widening of the Andes: An interplay between subduction dynamics and crustal wedge tectonics," *Earth-Science Rev.*, no. March 2020.
- [3] V. A. Ramos and A. Folguera, "Andean flat-slab subduction through time," *Geol. Soc. London, Spec. Publ.*, vol. 327, no. 1, pp. 31–54, 2009.
- [4] S. Zapata, E. R. Sobel, C. Del Papa, and J. Glodny, "Upper plate controls on the formation of broken foreland basins in the Andean retro-arc between 26 and 28  S: from Cretaceous rifting to Paleogene and Miocene broken foreland basins," *Geochemistry, Geophys. Geosystems*, vol. 21, no. 7, p. e2019GC008876, 2020.
- [5] V. A. Ramos, E. O. Cristallini, and D. J. Pe, "The Pampean flat-slab of the Central Andes," *J. South Am. Earth Sci.*, vol. 15, no. 1, pp. 6–8, 2002.
- [6] E. R. Sobel, G. E. Hilley, and M. R. Strecker, "Formation of internally drained contractional basins by aridity-limited bedrock incision," *J. Geophys. Res. Solid Earth*, vol. 108, no. B7, 2003.
- [7] P. Payrola et al., "Episodic out-of-sequence deformation promoted by Cenozoic fault reactivation in NW Argentina," *Tectonophysics*, vol. 776, p. 228276, 2020.
- [8] A. L. S. Goddard et al., "Reconstructing the thermal and exhumation history of the Sierras Pampeanas through low-temperature thermochronology : A case study from the Sierra de Velasco," no. Xx, pp. 1–17, 2018.

REVIEWERS' COMMENTS

Reviewer #2 (Remarks to the Author):

The authors have satisfactorily answered my comments and modified the manuscript accordingly. Thus, I consider that this manuscript should be published in its current form.